# Grazing Effects of Soil Fauna on White-Rot Fungi: Biomass, Enzyme Production and Litter Decomposition Ability

**DOI:** 10.3390/jof8040348

**Published:** 2022-03-28

**Authors:** Yunru Chen, Tingting Cao, Meiqi Lv, You Fang, Run Liu, Yunchao Luo, Chi Xu, Xingjun Tian

**Affiliations:** 1State Key Laboratory of Pharmaceutical Biotechnology, School of Life Sciences, Nanjing University, Nanjing 210023, China; cherry960502@163.com (Y.C.); ctting@smail.nju.edu.cn (T.C.); mg20300066@smail.nju.edu.cn (M.L.); youfang@smail.nju.edu.cn (Y.F.); recho24@126.com (R.L.); luoyunchao@outlook.com (Y.L.); 2Beijing Municipal Ecological and Environmental Monitoring Center, Beijing 100048, China; 3School of Ecological and Environmental Engineering, Qinghai University, Xining 810016, China; 4Co-Innovation Center for Sustainable Forestry in Southern China, Nanjing Forestry University, Nanjing 210042, China

**Keywords:** grazing, white-rot fungi, extracellular enzyme, fungal biomass, microcosm experiment

## Abstract

Soil invertebrates and microorganisms are two major drivers of litter decomposition. Even though the importance of invertebrates and microorganisms in biogeochemical soil cycles and soil food webs has been studied, the effects of invertebrates on fungi are not well understood compared to other organisms. In this work, we investigated the effects of soil invertebrates on fungi as a factor that cannot be ignored in the study of nutrient cycling. The result showed the grazing of isopods on white-rot fungi was transitive and persistent. The grazed fungi appeared “compensatory” growing. The biomass of fungi increased after grazing. The activities of enzymes associated with nutrient cycling were increased under grazing. The zymography images showed the enzyme hotspots and activities also increased significantly in the grazing area. The results suggest that invertebrate grazing can significantly increase the fungal biomass and enzyme activity, accelerating litter decomposition in the unreached grazer area. The grazing effects of invertebrate plays an important role in promoting the nutrient cycling of the forest ecosystem. We believe that this study will be a good reference related to showing the relationship between soil invertebrates, fungi and soil biogeochemical cycles.

## 1. Introduction

Litter decomposition is an important process driving the nutrient cycle in the terrestrial ecosystem [1], which is affected by many factors, such as microorganisms, soil invertebrates and climate [2,3].

Microorganisms are the main decomposers during the litter decomposition, which releases nutrients and provides energy for underground food webs and aboveground plant systems [4,5]. As the main driver of litter decomposition, the white-rot fungi can produce a broad spectrum of the extracellular enzyme [6,7,8]. The activity of the extracellular enzyme directly affects the rate of conversion of organic matter by microorganisms [9]. The enzyme is sensitive to environmental changes [10]. The activity of the enzyme was recognized as a sensitive indicator of litter decomposition [11].

Soil invertebrate is an important functional group of the soil ecosystem, which can directly or indirectly affect the decomposition process of litter [12]. They directly cause litter mass loss through feeding and mechanical crushing. They also indirectly affect litter decomposition by influencing microorganisms [13,14]. Litter broken by soil invertebrates can provide suitable niches for different kinds of microorganisms to settle down. Invertebrates could spread and carry fungal spores promoting resource colonization [15]. Some soil invertebrates could also directly feed on microorganisms [16,17]. Soil invertebrates interact with microorganisms, driving litter decomposition in terrestrial ecosystems [18].

Isopods are saprophytic invertebrates, a dominating invertebrate species in soil [19]. As macrofauna, isopods have a stronger impact on microbial communities and litter decomposition than other mesofauna and microfauna [20]. The macrofauna restricted the development of fungi hyphae [21]. Their large jaws and body allow them to graze the fungal hyphae, affecting the distribution of fungi.

Many studies emphasize invertebrate activities in litter decomposition and nutrient mineralization [12,20,22]. The effects of invertebrates or microorganisms on litter decomposition were often discussed. However, the study of the relationship between invertebrates and fungi is almost blank. Few studies have been conducted on the effects of invertebrate grazing on the growth and activity of individual decomposing fungi. Therefore, we designed an artificial microcosm to study the effects of invertebrate grazing on the enzyme production and litter decomposition ability of individual decomposing fungi. We hypothesized: (1) after grazing, the biomass of fungi in the grazing area decreased significantly, and the biomass of fungi growing on the litter in the decomposition area decreased correspondingly; (2) to obtain more resources and maintain their own growth, the extracellular enzyme secretion ability of decomposing fungi will be enhanced after grazing in both areas; (3) to counteract the negative effects of grazing on itself, the ability of fungi to decompose litter will increase significantly.

## 2. Materials and Methods

### 2.1. Collection and Preparation of Litter

Leaf litters were collected from a mixed forest dominated by *Quercus variabilis* and *Pinus massoniana* in Zijin Mountain (32°4′ N, 118°51′ E), Nanjing, China. The mountain has a subtropical monsoon climate with a mean annual temperature of 15.4 °C, and a maximum mean temperature of 28.2 °C in July and a minimum mean temperature of 1.9 °C in January. The rainy season is from June to July, and the average annual precipitation is 1106.5 mm. The soil is classified as slightly acidic Humic Cambisol with a pH of about 5.0 [9]. Four plots (2 m × 2 m), approximately 10 m apart, were chosen in the mixed forest. In October and November 2018, freshly fallen leaves of *P. massoniana* and *Q. variabilis* were collected at each of the four plots and then mixed each kind of litter separately.

The collected litters of *P. massoniana* and *Q. variabilis* were ground and sifted through 20 mesh. The litters were sterilized (autoclaving at 121 °C for three periods for 20 min) and oven-dried at 60 °C for 24 h to obtain a constant weight to be used for a subsequent study [23].

### 2.2. The Isolation and Culture of Decomposing Fungi

The decomposing fungi were isolated from forest soil in Zijin Mountain, where the leaf litters were collected. Soil samples were collected from the top layers (0–10 cm) of each plot. The collected soil samples were mixed and used to screen the decomposing fungi. The ground litter and soil were supplemented with sterile distilled water (10 mL water per 1 g of sample wet mass). Serial dilutions (10^−3^ to 10^−5^) were prepared. Potato dextrose agar (PDA) aniline blue lignin screening medium (6 g/L of potato extract, 20 g/L of glucose, 20 g/L of agar and 0.4 g/L aniline Blue) and Congo red cellulose screening medium (2 g/L of ammonium sulfate, 0.5 g/L of magnesium sulfate, 1 g/L of potassium dihydrogen phosphate, 2 g/L of sodium carboxymethyl cellulose, 0.1 g/L of sodium chloride, 0.4 g/L of Congo Red and 20 g/L of agar) [24] were used to screen fungi. The nalidixic acid (20 mg/L) was added to inhibit bacteria growth. A single colony from the plates was re-inoculated onto a screening medium. After separating and purifying three times on the screening medium, a strain with a strong degradation ability to lignin and cellulose was isolated. The fungus was identified as *Irpex lacteus* (Fr.) Fr., named as *Irpex lacteus* (Fr.) Fr.NJU-520, abbreviated as NJU-520 (Appendix A).

The NJU-520 was inoculated in 6 cm Petri dishes on PDA medium (1.5 g/L of potato extract, 5 g/L of glucose, 20 g/L of agar). All experimental equipment was sterilized (autoclaving at 121 °C for 20 min). The fungal cultures were inoculated in the center of Petri dishes. The decomposition experiment was conducted when the mycelium growth reached the edge of the Petri dish.

### 2.3. The Culturing of Grazers

*Procellio laevis* Latreille 1804 (Isopoda: Porcellionidae) is a dominant saprophytic fauna in the soil. It is widely distributed in the world, including experimental areas [25]. The isopods (*P. laevis* Latreille 1804) were purchased from a local shop (Isopod store, Jiangsu, China) and cultured with a mixture of sterilized *Q. variabilis* and *P. massoniana* litters (1:1) (fully moistened with sterile water). All containers were then stored in the dark at 25 °C. All the isopods were starved for 48 h in pots before being transferred into the experimental microcosms [9].

### 2.4. Construction of Artificial Microcosm

The artificial microcosm was constructed with a circular aluminum box (diameter = 10 cm and height = 6 cm). Ager (2%) was spread on the bottom of the microcosm for water conservation. The Petri dish was placed in the center of the microcosm, and an iron mesh (diameter: 6.4 cm, mesh size <1 mm)) (DehuaJinlongbin Ceramics Co., LTD., Quanzhou, China) was used to cover the Petri dish. The purpose was to confine isopods in the Petri dish and prevent them from contact with external litter directly. Two kinds of litter (2 g per microcosm) were separately placed between the net and Petri dish edge evenly. Fungi *I. lacteus* can extend to the decomposition area freely through the net. According to the survey, the number of isopods in 1 m^2^ of Zijin Mountain is about 401–581 [26]. Two isopods were added to each microcosm. The area inside the net where isopods graze was called the grazing area, and the area outside the net where litter was placed was called the decomposition area (Figure 1). According to the rate of isopods grazing fungi [27] and the change in enzyme activity [23], we set four sampling times (7 days, 21 days, 42 days and 70 days), two treatments (with or without isopods), two kinds of litter (*P. massoniana* and *Q. variabilis*), and blank controls, for a total of 144 treatments.

### 2.5. Determination of Fungal Biomass in Decomposition Area

Ergosterol is an important component of the fungal cell membrane. The biomass of fungi can be deduced simply by detecting the content of ergosterol. The method of determining ergosterol followed the reference of Ruzicka [28]. Sampling litters from the microcosm were placed in a 50 mL centrifuge tube at each sampling time, 10 mL methanol-ethanol mixture (4:1, *v*/*v* was added to the centrifuge tube, and frozen at 4 °C for 2 h. After freezing, 20 mL petroleum ether was added to each sample, immediately ultrasonicated while kept on ice, and the supernatant was absorbed into the new centrifuge tube. The supernatant was concentrated with a termovap sample concentrator (DC-12, ANPEL, shanghai, China) and dissolved with 1 mL methanol. Methanol was passed through a 0.22 μm filter membrane and then was measured by HPLC (P230II, Elite, Dalian, China). The chromatographic column was a reversed-phase column (4.5 mm × 200 mm), the mobile phase was methanol, the flow rate was 1 mL/min and the wavelength was 282 nm.

### 2.6. Enzyme Activity Determination in Decomposition Area

Determination of 1,4-*β*-glucosidase (BG, EC 3.2.1.21), cellobiohydrolase (CBH, EC 3.2.1.91), 1,4-*β*-N-acetylhexosaminidase (NAG, EC 3.2.1.52) and 1,4-*β*-xylosidase (BX, EC 3.2.1.37) followed the method of Jaroslav [29]. Acid phosphatase (AP, EC 3.1.3.1) activity was determined by the method of Allison Lab [30]. The substrates are listed in Table 1.

The litter sample was put into 8 mL 50 mmol/L sodium acetate buffer (pH 5.0) and extracted at 4 °C for 2 h on a shaker. Then, the sample was centrifuged and filtered to obtain the crude enzyme solution.

The activity of BG was assayed in microplates using *p*-nitrophenyl-*β*-d-glucoside. The reaction mixture contained 0.16 mL of 1.2 mM p-nitrophenyl-*β*-d-glucoside in 50 mM sodium acetate buffer (pH 5.0) and 0.04 mL sample. The reaction mixtures were incubated at 40 °C for 90–120 min. The reaction was stopped by adding 0.1 mL of 0.5 M sodium carbonate, and the absorbance was read at 400 nm (Safire microplate reader, TECAN, Männedorf, Switzerland). The activities of CBH, NAG and BX were assayed using the same method. The activity of AP was assayed in microplates using p-nitrophenyl-phosphate. The reaction mixture contained 0.15 mL of 5 mM p-nitrophenyl-phosphate in 50 mM sodium acetate buffer (pH 5.0) and 0.05 mL sample. The samples were incubated at 28 °C in the dark for 45 min, and 10 μL of 1 M NaOH was added to stop the reaction. The absorbance was read at a wavelength of 405 nm with a Safire microplate reader. (TECAN, Männedorf, Switzerland). One unit of enzyme activities was defined as the amount of enzyme releasing 1 μmol of p-nitrophenol per min.

### 2.7. Zymography in the Grazing Area

The activities of three enzymes related to the C, N and P cycle, 1,4-*β*-glucosidase (BG, EC 3.2.1.21), 1,4-*β*-*N*-acetylhexosaminidase (NAG, EC 3.2.1.52) and acid phosphatase (AP, EC 3.1.3.1), respectively, were determined using zymography according to Duyen T. T. Hoang [31].

The substrates (4-Methylumbelliferyl-*β*-d-glucoside MUF-G, 4-Methylumbelliferyl-phosphate MUF-P, 4-Methylumbelliferyl-*N*-acetyl-*β*-d-glucosaminide MUF-N) were separately dissolved to a concentration of 12 mM in MES (C_6_H_13_NO_4_SNa_0.5_) buffer. Polyamide membrane filters (diameter 20 cm, pore size 0.45 mm—Tao Yuan, China) were cut into pieces of the required size and soaked in the prepared substrate solutions. The membranes were applied directly to the surface of the grazing area. After 60 min of incubation, the membranes were placed under ultraviolet (UV) illumination with an excitation wavelength of 355 nm and an emission wavelength of 460 nm in a light-proof room. In order to maintain the constant conditions for all samples, the distance between the UV light resource, the camera (D3400, Nikon, Tokyo, Japan) and the samples was fixed [32].

To quantify the zymogram images, we calibrated against standards that related the enzyme activities to the gray value projected onto the zymograms. The calibration function for each enzyme was obtained by zymography of 2 cm^2^ membranes soaked in solutions of 4-methylumbelliferone (MUF) at concentrations of 0.05, 0.06, 0.07, 0.08, 0.09 and 0.1 μM. The amount of MUF per area basis was calculated from the volume of solution taken up by the membrane and the membrane size. The calibration membranes were imaged under UV light and analyzed in the same way as the samples. The standard curve of the correlation between enzyme activity and gray value was obtained, R^2^ to 0.99 (Appendix A).

### 2.8. Physical and Chemical Properties of Litter

The collected litters were dried at 60 °C for 48 h, and 5 mg dry litters were put into an element analyzer (Elemental Vario MICRO, Langenselbold, Germany) to determine the concentration of total carbon and total nitrogen. The determination of pH was carried out with reference to the method of Fioretto [33].

### 2.9. Statistical Analysis

All statistical tests were performed using SPSS (version 22.0, SPSS Inc., Chicago, IL, USA). Data were checked for deviations from normality and homogeneity of variance before analysis. Analysis of variance (ANOVA) and Tukey’s HSD (honest significant difference) test was applied to assess differences among treatments. One way-ANOVA was used to analyze the effects of fungal biomass, carbon and nitrogen content and enzyme activity.

In order to obtain quantitative information, we processed the zymograms using the image processing toolbox in Matlab 2019a (MathWorks, Natick, MA, USA. The picture taken by the camera was converted into a 16-bit grayscale image and the standard curve to convert the grayscale value into enzyme activity [32]. The unwanted noise caused by the light variation and the camera was corrected [32]. In order to illustrate the results, values from the greyscale images were converted to in color [32]. Based on the previous studies, we defined enzymatic hotspots as gray values exceeding 25% of the mean gray value of the entire image [34]. The hotspot areas were then presented as a percentage of the total soil surface area [34]. All graphs were drawn using Origin 2017 (OriginLab, Northampton, MA, USA).

## 3. Results

### 3.1. Fungal Biomass in the Decomposition Area

The fungal mycelia extended from grazing area to decomposition area after 7 days incubation. The fungal biomass of the litter in the decomposition area increased gradually during 70 days of incubation. The fungal biomass of *Quercus variabilis* litter increased rapidly in the first 7 days (Figure 2). The fungal biomass of the litter was higher under grazing than non-grazing. Subsequently, the fungal biomass increased continuously, and both treatments reached the same level on the 21st day. The fungal biomass increased continuously in the grazing treatment but decreased in the non-grazing treatment on the 42nd day. After 70 days, the highest fungal biomass of *Q. variabilis* litter was found in the grazing treatment (40.04 μg/g), which was higher than that in the non-grazing treatment (15.49 μg/g).

The fungi colonized of *Pinus massoniana* litter also showed an increasing trend with time. At first, the increase in the non-grazing treatment was higher than that in the grazing treatment. Then, they reached the same level on the 42nd day. Then the fungal biomass in the grazing treatment continued to increase. After 70 days, the fungal biomass of *P. massoniana* litter in the grazing treatment (13.07 ± 4.96 μg/g) was higher than that in the non-grazing treatment (8.36 ± 1.69 μg/g). The above results suggest that isopods grazing accelerated the colonization of fungi on the litter, especially on *Q. variabilis* litter.

### 3.2. Enzyme Activity in Decomposition Area

In the decomposition area of two litters, all enzyme activities increased continuously within 70 days with grazing. Enzyme activities were higher in the grazing treatment than that in the non-grazing treatment (Figure 3).

During the 70 days, the activity of AP of *Q. variabilis* litter in the decomposition area under the grazing treatment was higher than the non-grazing treatment. On the 70th day, the activity of AP under the grazing treatment was 2.48 times higher than non-grazing treatment in *Q. variabilis* litter (*p* < 0.05). The activity of BG in *Q. variabilis* litter showed an upward trend during the whole process. On the 42nd day of decomposition, the activity of BG under grazing treatment was 2.05 times higher than non-grazing control in *Q. variabilis* litter (*p* < 0.05, Figure 3). The activity of CBH of *Q. variabilis* litter showed an increasing trend with the whole period. On the 70th day, the activity of CBH under the grazing treatment was 0.78 times higher than the non-grazing control in the *Q. variabilis* litter. At the initial stage of decomposition, the activity of BX on the *Q. variabilis* litter was not detected. During the decomposition, the activity of BX was detected in the litter under grazing treatment in the early stage. After the 42nd day, a significant difference was found between grazing treatment and non-grazing treatment of BX (*p* < 0.05).

The activity of AP of *P. massoniana* litter under the grazing treatment was continuously higher than that of the non-grazing treatment during the 70 days. On the 70th day, the activity of AP under the grazing treatment was 60 times higher than that of the non-grazing treatment (*p* < 0.05). The activity of NAG under the grazing treatment increased rapidly within 70 days, but the non-grazing treatment group was continuously at a low level (*p* < 0.05). The activity of BG in *P. massoniana* litter showed an upward trend, and the activity of BG in the grazing treatment was continuously higher than that in the non-grazing treatment during the 70 days. On the 42nd day, the activity of BG under the grazing treatment increased by 3.43 times compared with that of the non-grazing treatment (*p* < 0.05). The activity of CBH in *P. massoniana* litter continuously increased with time. On the 70th day, the activity of CBH under the grazing treatment was 2.52 times higher than that in the non-grazing treatment. At the initial stage of decomposition, no activity of BX was detected in *P. massoniana* litter; then, the activity of BX was first detected in the litter under grazing treatment. However, the activity of BX was not detected in *P. massoniana* litter in the non-grazing treatment for 70 days.

### 3.3. The Activity and Hotspot of Enzyme in Grazing Area

The zymogram images of acid phosphatase, *β*-N-acetylhexosaminidase and *β*-1,4-glucosidase in the grazing area at 7 days, 21 days, 42 days and 70 days are shown in Figure 4 and Figure 5. The color of zymogram images could be divided into red, orange, yellow, green, blue and dark blue, which represents the enzyme activity from high to low and even no activity separately.

On the 7th day, grazing did not lead to a significant change in fungal biomass (as the picture shows), but the activity of AP in the grazing area increased significantly (*p* < 0.05, Table 2). Then, the fungal biomass decreased under the grazing treatments in the grazing area. On the 70th day, almost no fungus could be found in the grazing area; however, the enzyme activity did not disappear. The activity of NAG in the grazing area under grazing treatment was continuously higher than that without grazing and reached a significant level on the 42nd day (*p* < 0.05, Table 2). The change in *β*-1,4-glucosidase activity was similar to that of *β*-N-acetylhexosaminidase activity. On the 42nd day, the enzyme activity in the grazing area with isopods grazing was significantly higher than that without grazing (*p* < 0.05, Table 2). The hotspots of the zymogram images in the grazing area with isopods were more than that without isopods (Appendix A). The above results suggest whether the external litter was *Q. variabilis* or *P. massoniana*; grazing increased the activity of the enzyme in the grazing area. The activity of the enzyme in the treatment with isopods grazing was significantly higher than that in the non-grazing treatment (*p* < 0.05).

### 3.4. Litter Decomposition

The decomposition rate of *Q. variabilis* was higher than that of *P. massoniana* (*p* < 0.05, Figure 6). Grazing had no significant effect on litter decomposition. On the 70th day of decomposition, the C/N of the two kinds of litter grazed by isopods was the lowest, but the difference was not significant compared with the treatment without grazing.

It is the correlations among decomposition time, grazing treatment, litter mass loss, C content, N content, pH and five enzyme activities during 70 days (Figure 7). The isopods grazing was significantly positively correlated with BG, AP, BX, NAG, CBH and pH (*p* < 0.05). The mass loss of litter positively correlated with the activities of BG, AP and CBH, and N content (*p* < 0.01), but there was no significant correlation with isopods grazing.

## 4. Discussion

We hypothesized that the biomass of fungi decreased both in the grazing area and the decomposition area. However, the results were not exactly in line with the hypothesis. Our result showed the fungal biomass decreased significantly in the grazing area but increased in the decomposition area. The previous study confirmed that large jaws and body size of macrofauna enabled severing thick cords [20]. The grazing of isopods causes a large amount of hyphal damage [20]. Therefore, fungal biomass decreased in the grazing area. However, our work showed that the fungal biomass increased in the decomposition area under the grazing of isopods in the grazing area. We found that fungi could develop a “compensatory” growth strategy under grazing. Previous studies reported that *Hypholoma fasciculare* DD3 and *Phanerochaete velutina* could use dense mycelium to encounter resources on a small spatial scale [20,35]. These two species showed a higher growth rate during the invertebrate invasion. Fungi actively increase their biomass to search for new nutrients and produce more hyphae after being threatened to offset the damage caused by the threat [36]. Under grazing, the increase in fungal biomass of *I. lacteus* in the decomposition area indicates that decomposing fungi have the ability to resist the negative effects of grazing.

We hypothesized the extracellular enzyme secretion ability of decomposing fungi would be enhanced after grazing in both areas to obtain more resources. In line with the second hypothesis, the activities of the enzyme increased after grazing in both areas. A previous study showed that the shifts in extracellular enzyme activity of fungi were related to their growth and activity during invertebrate invasion [37]. Similar results reported that grazing increased the production of extracellular enzymes in non-grazing areas of mycelium [37]. The reason for the increase may be the mechanical damage caused by grazing may lead to the leakage of enzymes in fungal cells or the release of cellular contents, thus increasing the production of microbial enzymes to digest substrates [37]. The increased enzyme activities in both areas showed that grazing effects were transitive and persistent. The activity of the enzyme was also related to the growth strategy of fungi [15,23]. Similar work reported that under the fungi biomass increased, adopted a “compensatory” growth strategy, the enzyme activity of the fungi generally increased [23]. Continuous changes in the activities of enzymes involved in cellulose decomposition and phosphorus uptake during grazing indicate that invertebrates may have a special impact on the cycle of carbon and phosphorus in forest ecosystems. The feces and activities of isopods may promote the production of hotspots of enzyme activity [9]. Zymography pictures also showed the hotspots of enzyme activity with isopods were higher than that without isopods. Therefore, it can be speculated that invertebrate plays a positive role in maintaining the enzyme activity secreted by microorganisms during the decomposition of forest litter.

In contrast to the third hypothesis, litter decomposition was accelerated but not significantly under grazing. Similar works reported that the interaction between fungi and soil fauna significantly increased wood decay during high-intensity grazing [20,37]. Grazing directly led to an increase in fungal enzyme activity and indirectly increased wood decomposition [20,37]. Fungi can adjust the negative effects of grazing by increasing enzyme production and nutrient absorption [37]. These studies proved that soil invertebrates could stimulate fungal-mediated nutrient mineralization and decomposition [38]. However, our study showed no significant decrease in litter decomposition after grazing. It is possible because the decomposition period was short, the overall enzyme activities and interference intensity were lower. Coulis pointed out that the conversion of litter into feces by macrofauna and their interaction with microorganisms did not enhance carbon mineralization in the short term [39]. In this study, the fungi in the grazing area disappeared completely within 70 days. If a larger scale experiment is conducted, the significant acceleration on litter decomposition may gradually appear.

## 5. Conclusions

This study suggests the other way of invertebrates to promote litter decomposition and accelerate the forest nutrient cycle in addition to mechanical crushing. Invertebrates can directly increase the growth and enzyme activity of fungi through grazing. White-rot fungi adopt a “compensatory” growth strategy in the face of fauna grazing. These grazing effects in the grazing area can be transmitted through hyphae, increasing the growth and extracellular enzyme activities of fungi in the non-grazing area (decomposition area). The decomposition of fungi was affected by the litter species. In conclusion, invertebrate grazing is a crucial pathway of the carbon dynamics and nutrient cycling driven by decomposing fungi in the forests.

## Figures and Tables

**Figure 1 jof-08-00348-f001:**
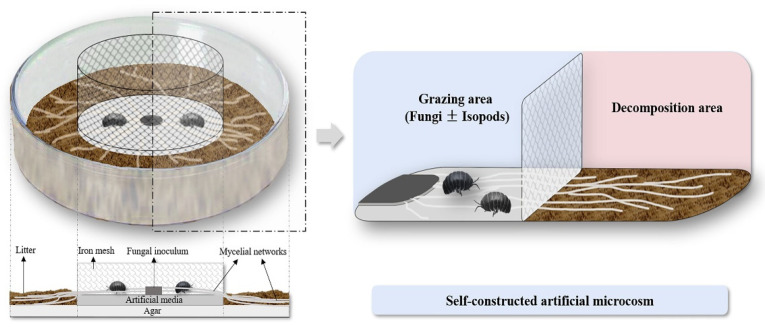
The artificial microcosm.

**Figure 2 jof-08-00348-f002:**
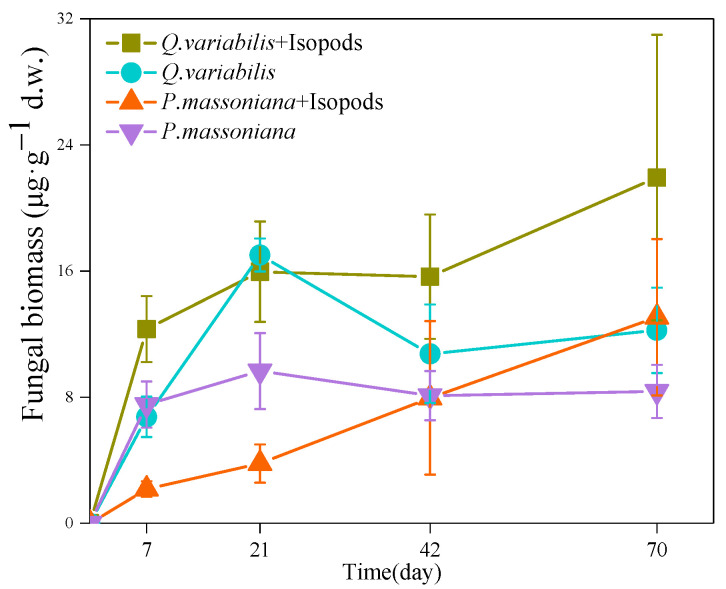
Fungal biomass changes in the decomposition area under different treatment during 70 days incubation.

**Figure 3 jof-08-00348-f003:**
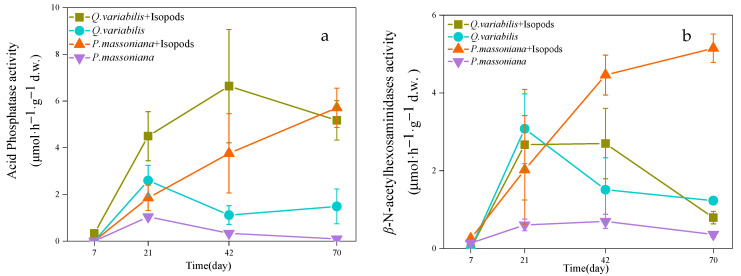
Changes in the activities of acid phosphatase (**a**), *β*-*N*-acetylhexosaminidases (**b**), *β*-1,4-glucosidase (**c**), cellobiohydrolases (**d**), 1,4-*β*-xylosidase (**e**) of litter in the decomposition area under different treatments during 70 days incubation.

**Figure 4 jof-08-00348-f004:**
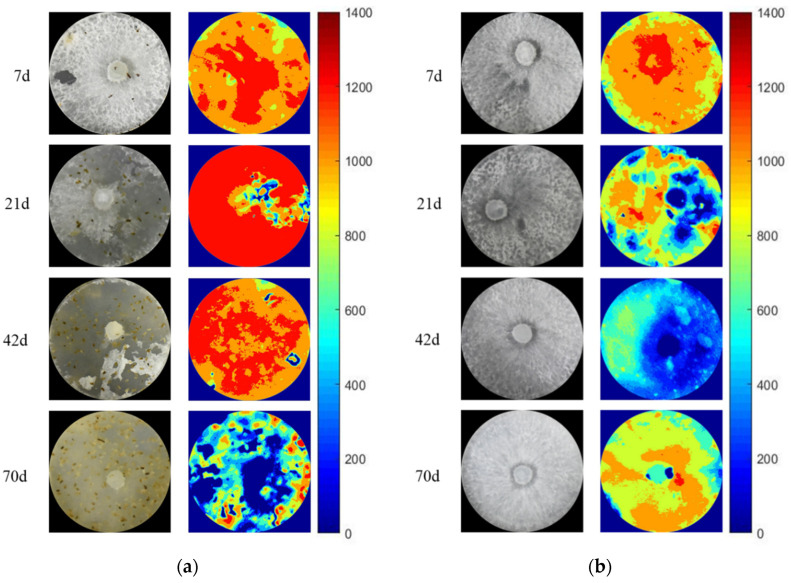
Spatial distribution of enzyme activities of grazing areas (External litter: *Q. variabilis*): (**a**,**b**) acid phosphatase, (**c**,**d**) *β*-*N*-acetylhexosaminidase, (**e**,**f**) *β*-1,4-glucosidase. (**a**,**c**,**e**) under isopods grazing; (**b**,**d**,**f**) without isopods grazing. The side color map is proportional to the enzyme activities (pmol·cm^−2^·h^−1^).

**Figure 5 jof-08-00348-f005:**
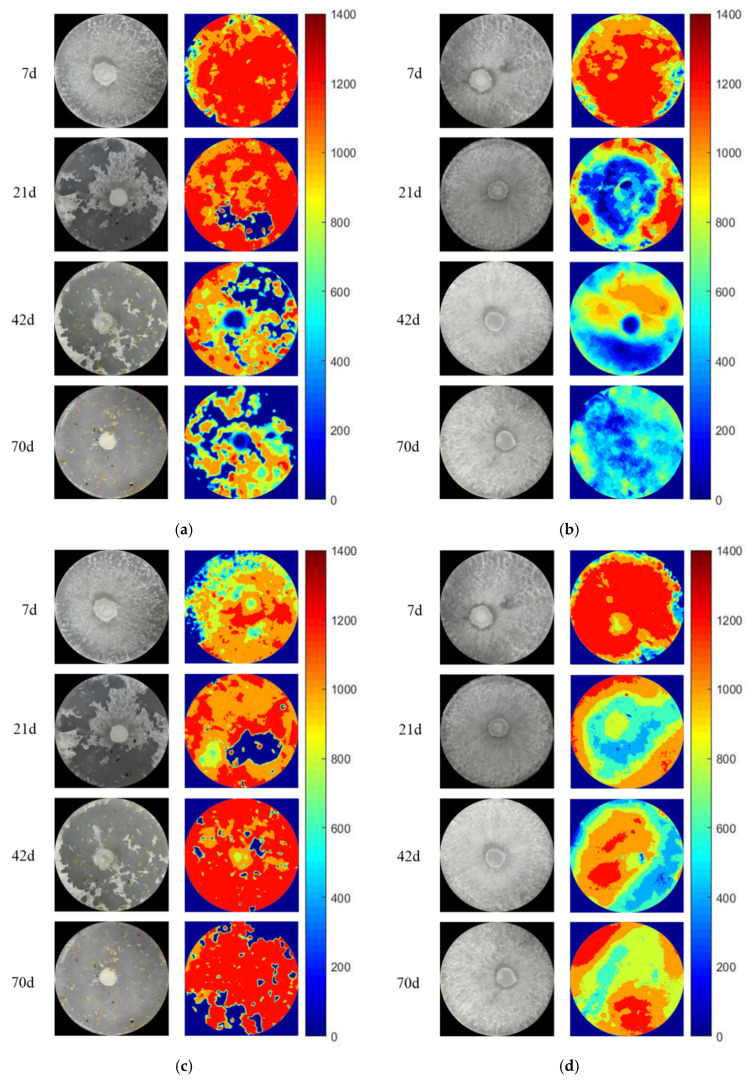
Spatial distribution of enzyme activities of grazing area (External litter: *P. massoniana*): (**a**,**b**) acid phosphatase, (**c**,**d**) *β*-*N*-acetylhexosaminidase, (**e**,**f**) *β*-1,4-glucosidase. (**a**,**c**,**e**) under isopods grazing; (**b**,**d**,**f**) without isopods grazing. Side color maps are proportional to the enzyme activities (pmol·cm^−2^·h^−1^).

**Figure 6 jof-08-00348-f006:**
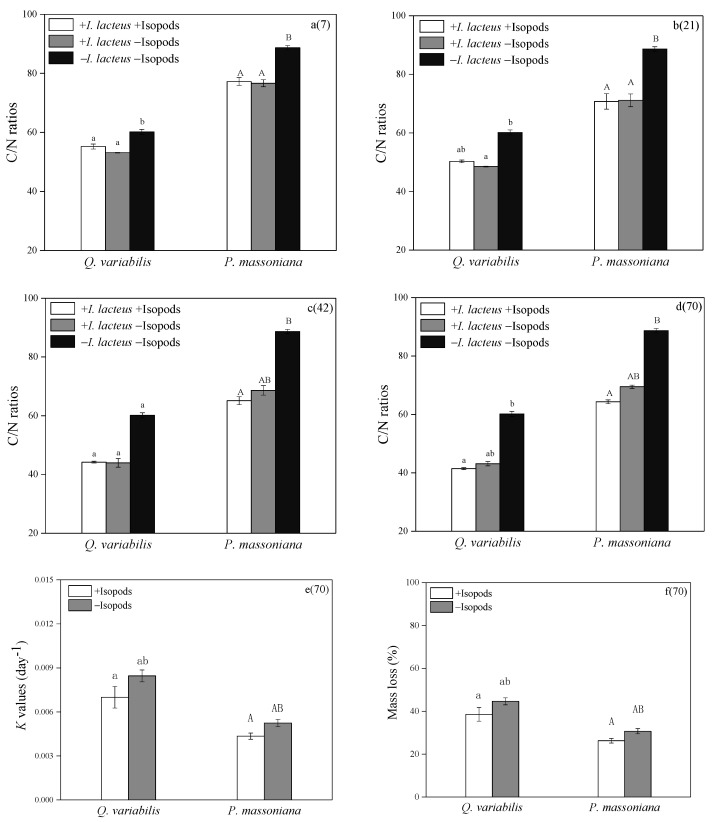
Physiochemical properties of *Q. variabilis* and *P. massoniana* on 7 days (**a**), 21 days (**b**), 42 days (**c**), 70 days (**d**), decomposition rates (mean k value, day^−1^) (**e**) and mass loss (**f**) of the two leaf litters during 70 days of decomposition.

**Figure 7 jof-08-00348-f007:**
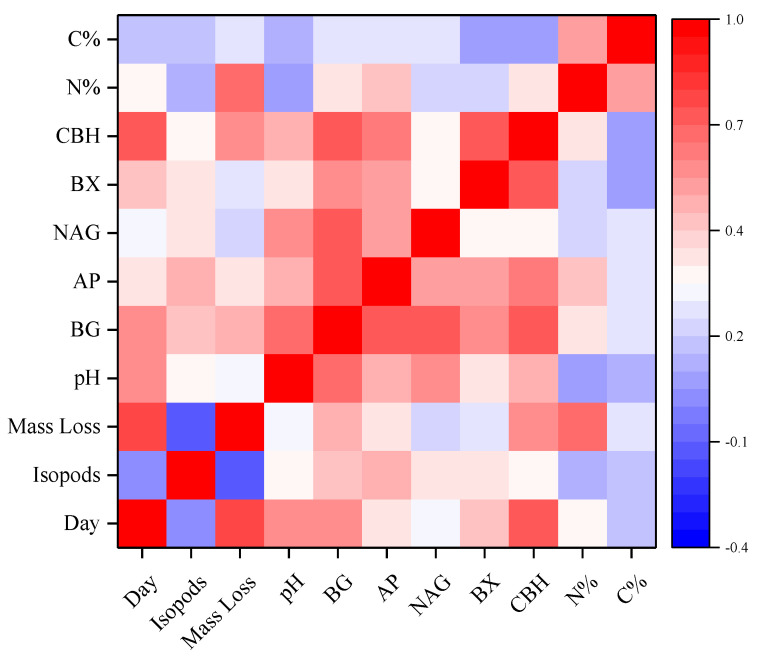
Heatmap of correlation analysis among factors. The color diagram on the right represents the level of correlation between factors.

**Table 1 jof-08-00348-t001:** Enzymes and substrates related to carbon (C), nitrogen (N), and phosphorus (P) cycles.

Enzyme	EC	Substrate
*β*-1,4-glucosidase	3.2.1.21	*p*-nitrophenyl-*β*-d-glucoside
*β*-*N*-acetylhexosaminidases	3.2.1.52	*p*-nitrophenyl-N-acetyl-*β*-d-glucosaminide
Acid phosphatase	3.1.3.1	*p*-nitrophenyl-phosphate
Cellobiohydrolases	3.2.1.91	*p*-nitrophenyl-*β*-d-cellobioside
1,4-*β*-xylosidase	3.2.1.37	*p*-nitrophenyl-*β*-d-xyloside

**Table 2 jof-08-00348-t002:** Enzyme activity measured by zymography at 7 days, 21 days, 42 days and 70 days in the grazing area (pmol·cm^−2^·h^−1^).

Enzyme	Treatment		Day			
			7	21	42	70
Acid phosphatase	*Q. variabilis*	+Isopods	612.37 ± 5.00 ^b^	620.29 ± 86.08 ^c^	608.31 ± 17.69 ^b^	—
		−Isopods	531.81 ± 23.47 ^a^	316.33 ± 65.16 ^a,b^	—	409.07 ± 41.06 ^b^
	*P. massoniana*	+Isopods	635.99 ± 6.20 ^b^	566.18 ± 70.85 ^b,c^	345.64 ± 44.68 ^a^	195.40 ± 82.38 ^a,b^
		−Isopods	613.77 ± 2.04 ^b^	215.82 ± 86.03 ^a^	117.81 ± 63.20 ^a^	108.32 ± 18.19 ^a^
*β*-*N*-acetylhexosaminidases	*Q. variabilis*	+Isopods	488.86 ± 83.05 ^a^	656.25 ± 72.65 ^a^	604.36 ± 24.57 ^b^	556.51 ± 75.28 ^a^
		−Isopods	416.01 ± 59.80 ^a^	520.60 ± 52.14 ^a^	455.76 ± 15.42 ^a^	537.68 ± 14.79 ^a^
	*P. massoniana*	+Isopods	503.75 ± 62.00 ^a^	478.25 ± 47.24 ^a^	656.23 ± 16.41 ^b^	477.55 ± 41.14 ^a^
		−Isopods	521.59 ± 102.21 ^a^	322.94 ± 112.26 ^a^	394.83 ± 9.30 ^a^	472.68 ± 29.62 ^a^
*β*-1,4-glucosidase	*Q. variabilis*	+Isopods	434.28 ± 24.79 ^a^	648.54 ± 64.08 ^a^	581.17 ± 18.33 ^b^	510.62 ± 85.87 ^a^
		−Isopods	403.62 ± 10.33 ^a^	534.39 ± 68.87 ^a^	341.12 ± 18.27 ^a^	460.75 ± 40.11 ^a^
	*P. massoniana*	+Isopods	470.92 ± 7.33 ^a^	634.28 ± 70.47 ^a^	544.82 ± 56.65 ^b^	385.41 ± 75.26 ^a^
		−Isopods	417.36 ± 44.33 ^a^	414.96 ± 98.73 ^a^	237.66 ± 59.17 ^a^	253.48 ± 84.15 ^a^

+Isopods: with isopods; −Isopods: without isopods. Data with different superscript letters in a transverse row are significantly different (*p* < 0.05, *n* = 3).

## Data Availability

Not applicable.

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
