# Peer review of "Grazing Effects of Soil Fauna on White-Rot Fungi: Biomass, Enzyme Production and Litter Decomposition Ability"

_jof, 2022, doi:10.3390/jof8040348_

Round 1
Reviewer 1 Report
I did several remarks in the attached reviewed pdf.
Olaf Schmidt

Reviewer 2 Report
The manuscript on the Grazing effects of soil fauna on white rot fungi: biomass and enzyme production and litter decomposition ability is of interest regarding the grazing. The research is interesting, quite novel and but paper is rather not well written and presented.
Abstract is not well written; it is only a mere conscript of the study. Better would be to give some introduction followed by the gap in knowledge, hypothesis, general results and then conclusion. The abstract is the only part of the paper that the vast majority of readers see. Therefore, it is critically important for authors to ensure that their enthusiasm or bias does not mislead the reader.
The introduction resembles that of a review article and not that of a research article. What’s the gap of knowledge? Which is the scope of the manuscript? What hypothesis have been made? The introduction should be revised accordingly.
Experimental section:. A more succinic yet complete writing should be done. Moreover the author state that a statistical analysis has been made. I believe that the authors should give more details about the analysis performed.
Discussion: Again this part requires re-writing. The results should be more accurately being related with previous studies and possible interpretation of the findings should be more clearly stated.
Check the below references for the improvement of your introduction and discussion portion.
Fahad, S., Sonmez, O., Saud, S., Wang, D., Wu, C., Adnan, M., Turan, V. (Eds.), 2021b. Climate change and plants: biodiversity, growth and interactions, First edition. ed, Footprints of climate variability on plant diversity. CRC Press, Boca Raton.
Fahad S, Hasanuzzaman M, Alam M, Ullah H, Saeed M, Ali Khan I, Adnan M. (Eds.) (2020) Environment, Climate, Plant and Vegetation Growth. Springer Nature Switzerland AG 2020. DOI: https://doi.org/10.1007/978-3-030-49732-3
Round 2
Reviewer 2 Report
Accepted as it stands